# Prevalence of Anal High-Risk Human Papilloma Virus Infection and Abnormal Anal Cytology among Women Living with HIV

**DOI:** 10.3390/jpm12111778

**Published:** 2022-10-28

**Authors:** Leticia Muñoz-Hernando, Reyes Oliver-Pérez, Victoria Bravo-Violeta, Alejandro Olloqui, Belen Parte-Izquierdo, Cristina Almansa-González, Rocio Bermejo-Martinez, Ana Belen Bolivar-De Miguel, Alvaro Diez, Jose Miguel Seoane-Ruiz, Lucia Parrilla-Rubio, Alvaro Tejerizo-García

**Affiliations:** 1Colposcopy and Cervical Pathology Unit, Department of Obstetrics and Gynaecology, University Hospital 12 de Octubre, 28041 Madrid, Spain; 2Instituto de Investigacion Sanitaria Hospital 12 de Octubre (imas12), Universidad Complutense de Madrid, 28041 Madrid, Spain; 3Gynaecology Oncology Unit, Department of Obstetrics and Gynaecology, University Hospital 12 de Octubre, Avda, Córdoba s/n, 28041 Madrid, Spain; 4Department of Pathology, University Hospital 12 de Octubre, Avda, Andalucía s/n, 28041 Madrid, Spain

**Keywords:** human papillomavirus, HPV, screening, anal, human immunodeficiency virus, HIV, women

## Abstract

Background: Women living with human immunodeficiency virus (HIV), WLWHs, are at high risk of developing anal cancer associated with high-risk human papilloma virus infection (HR-HPV). We analyzed the prevalence of anal HR-HPV infection and abnormal anal cytology in a cohort of WLWHs and assessed the risk factors for anal HR-HPV infection. Methods: We present a single-center, observational cross-sectional study. WLWHs who underwent anal cytology and anal human papilloma virus (HPV) testing were selected. High-resolution anoscopy was performed in cases of abnormal anal cytology. All suspicious lesions were biopsied. A univariate and multivariate logistic regression model was used to analyze risk factors for abnormal anal screening. The results are presented as odds ratios (ORs) and 95% confidence intervals (CIs). Results: In total, 400 WLWHs were studied. Of them, 334 met the eligibility criteria and were enrolled in the study. Abnormal anal cytology was detected in 39.5% of patients, and anal HR-HPV in 40.1%, with HPV 16 in 33 (26.6%) of them. Concomitant HR-HPV cervical infection was the only independent risk factor for HR-HPV anal infection (OR 1.67 95% CI, *p* < 0.001). Conclusions: WLWHs have a high prevalence of HR-HPV anal infection and anal cytologic abnormalities. HR-HPV cervical infection is the main predictor of HR-HPV anal infection.

## 1. Introduction

Anal cancer is an uncommon disease in the general population, with an estimated incidence of 1–2 per 100,000 person-years (py) but increasing in recent years in Western countries [1,2]. However, the incidence of anal cancer is substantially higher in people living with human immunodeficiency virus (PLWH, HIV) [3]. Women living with HIV (WLWHs) are at particularly high risk of developing anal cancer, with an estimated incidence of 22 per 100,000 py [4].

Human papilloma virus (HPV) infection is associated with cervical and anogenital cancers, as well as skin and mucosal lesions. Twelve genotypes have been identified as high-risk types due to their oncogenic characteristics and association with cancer and precursor lesions, whereas types 6, 11, and others, classified as low-risk types, are associated with benign diseases such as genital warts [5].

Around 90% of anal cancers are attributable to a high-risk human papilloma virus (HR-HPV) infection, with Human HPV 16 being the most frequently detected genotype [6]. The relationship between HIV and HPV pathogenesis has been studied. Due to immunosuppression associated with HIV infection, PLWHs have a higher risk of HPV acquisition, lower clearance, and higher risk of precancerous lesions and cancer than immunocompetent people [7,8].

As for cervical cancer, anal cancer is preceded by high-grade squamous intraepithelial lesions (HSILs) [9,10,11]. Recently, the Anal Cancer HSIL Outcomes (ANCHOR) study concluded that PLWH who underwent early identification and treatment for anal HSIL had a lower rate of progression to anal cancer [12]. However, there have been no controlled studies to determine whether anal cancer screening programs are effective. Screening recommendations for PLWH are based on experts’ opinions with a lack of consensus. The strategies simulate cervical cancer screening models, performing annual cytology with/without HPV molecular testing, and high-resolution anoscopy (HRA) in cases of abnormal results [13,14,15,16].

The aim of this study was to analyze the prevalence of anal HR-HPV infection and the prevalence of abnormal anal cytology in a cohort of WLWHs. Additionally, we aimed to assess the risk factors for anal HR-HPV infection.

## 2. Materials and Methods

We present a single-center, observational, cross-sectional study performed from January 2011 to June 2022 at the Colposcopy and Cervical Pathology Unit of Hospital Universitario 12 de Octubre, Madrid, Spain. WLWHs who were referred for cervical cancer screening or follow-up for a history of cervical H-SIL and underwent concomitant anal cytology and anal HPV testing were enrolled. Patients were excluded who were under 18 years of age with a previous diagnosis of anal cancer and/or in whom neither anal cytology nor anal HPV testing was performed.

Clinic and pathologic data were retrospectively obtained from sourced documentation in the electronic medical records (clinic notes and pathologic reports). The study was approved by our institution’s ethics committee (Instituto de Investigacion Sanitaria Hospital 12 de Octubre [imas12].) (N°CEI: 21/710), which waived the necessity for informed consent from the patients due to the retrospective nature of the study. All researchers involved agreed to treat the data confidentially in accordance with the General Data Protection Regulation and the Declaration of Helsinki [17].

### 2.1. Sampling Procedures

Anal and cervical cytology and anal and vaginal HPV tests were performed on all patients at the same visit.

All anal cytology specimens were collected using a moistened cotton swab. The swab was inserted blindly 2–4 cm deep into the anal canal and rotated 360° for 60 s to ensure that the entire canal was sampled. All cervical cytology specimens were collected using a broom, inserting the central bristles into the endocervix with the outer bristles in contact with the ectocervix, and rotating the broom five turns in the same direction. The swab and the broom were then each placed in a separate preservative solution (PreservCyt; Cytyc Corp., Boxborough, MA, USA) for thin-layer preparation (ThinPrep). Specimens of the anal and cervical cytology were processed on a ThinPrep (Hologic) processor and interpreted by a cytopathologist. The results are reported using Bethesda System terminology (negative for intraepithelial lesion or malignancy (NILM); atypical squamous cells of undetermined significance (ASCUS); atypical squamous cells, cannot rule out high-grade squamous intraepithelial lesion (ASC-H); low-grade squamous intraepithelial lesion (LSIL); and high-grade squamous intraepithelial lesion (HSIL)) [9].

Polymerase chain reaction assays for HPV DNA detection were performed with a Clart^®^ HPV system (Genomica SAU, Madrid, Spain). This system detects 49 different genotypes of HPV, including the most prevalent high- and low-risk genotypes [18].

### 2.2. Management of Abnormal Results

In patients with abnormal anal cytology, HRA was performed using a standard colposcope (Olympus OCS5-DCAD2; Olympus, Center Valley, PA, USA). Prior to examination, 5% acetic acid solution was applied into the anal canal. Findings were classified according to standard colposcopy nomenclature [19]. All suspicious lesions were biopsied. The procedures were performed by a gynecologist specialized in colposcopy and cervical pathology with at least five years of experience.

Biopsies were formalin-fixed and paraffin-embedded, sectioned into 4 μm slides and stained with hematoxylin–eosin for routine histological study. Biopsies were classified as benign, LSIL (anal intraepithelial neoplasia type 1), or HSIL (anal intraepithelial neoplasia types 2/3) [11]. All cytologic and histologic exams were performed by specialized HPV-related disease pathologists. Both HRA and anal biopsies were carried out without anesthesia.

### 2.3. Statistical Analysis

All patients were evaluated for their basal characteristics. Additional information collected for this group of women included the time of HIV infection, HIV viral load, CD4 count, or time of antiretroviral therapy at the time of anal cytology. Outcome data were recorded in a common database created on the Research Electronic Data Capture (REDCap) tool hosted at the “imas12” research institute [20].

Categorical variables are expressed as relative and absolute frequencies and quantitative data as the mean (standard deviation, SD) when a normal distribution could be assumed, or as the median and interquartile range when it was not possible. Normality was tested using the Shapiro–Wilk test. Statistical hypothesis tests were performed to assess the associations between stratifying and explanatory variables. For categorical variables, Chi-squared tests were used; if the expected frequency in any of the observations was lower than 5, Fisher’s exact test was used instead. As for numerical variables, when normality was assumed, Student’s t-tests for independent samples were performed. When normality could not be assumed, non-parametric Mann–Whitney U tests were used instead.

To study the association between anal HPV infection and other risk-related parameters, univariate logistical regression models were applied. To avoid possible confounding factors, multivariate logistic regression analysis was performed. The results are summarized by odds ratios (ORs) and their corresponding 95% confidence intervals (CIs). For all the analyses, a two-sided *p*-value of less than 0.05 was considered statistically significant. All the statistical analysis was performed with Stata/IC 13.0 for Windows.

## 3. Results

In total, 400 WLWHs were studied. Of them, 334 patients met the eligibility criteria and were enrolled in the study (Figure 1). Clinical and HIV infection features of all patients are summarized in Table 1 and Table 2. The mean age was 47.1 years, and 76.8% of patients had at least 10 years of HIV infection.

As shown in Table 3, 132 (39.5%) patients had an abnormal anal cytology and 134 (40.1%) had an HR-HPV anal infection, with HPV 16 in 33 (24.6%) of them. Among women with abnormal anal cytology, 46.9% underwent HRA. In total, 62 (18.6%) HRAs and 12 (3.6%) anal biopsies were performed. Two cases of anal HSIL were diagnosed with this protocol. Concomitant HR-HPV cervical infection was detected in 28% of patients, 24.5% of them exhibiting HPV genotype 16.

Table 4 shows a comparative analysis of patients’ characteristics according to abnormal anal screening. For the analysis, abnormal cytology was considered as any result equal to or greater than ASCUS [9]. There were no statistically significant differences between patients with negative and abnormal cytology regarding the parameters analyzed. HR-HPV anal infection increased with longer HIV infection (*p* < 0.001), higher viral load level (*p* = 0.037), lower CD4+ level (*p* = 0.002), shorter antiretroviral treatment time (*p* = 0.017), and in patients with HR- HPV cervical infection (*p* < 0.001).

Multivariate analysis revealed that HR-HPV cervical infection (OR 5.1; 95% CI 2.8; 9.2, *p* < 0.001), remained an independent risk factor for HR-HPV anal infection. HIV infection time, HIV viral load level, CD4+ level, and time of antiretroviral therapy were not independent risk factors, as shown in Table 5.

**Table 1 jpm-12-01778-t001:** Clinical and HIV infection features.

Characteristics	Values
Age, years	
Mean (SD)	47.1 (8.6)
Median (IQR)	48.0 (42.0; 53.0)
Age, years	
<35	38 (11.4)
35–50	171 (51.2)
>50	125 (37.4)
Postmenopausal NR: 11	141 (43.7)
Parity	
NR: 41	
Mean (SD)	1.5 (1.1)
Median (IQR)	1.0 (1.0; 2.0)
HIV infection time, years	
≤2	34 (10.9)
3–9	38 (12.3)
≥10 NR: 24	238 (76.8)
HIV viral load level, copies/mL	
<50	287 (89.1)
50–400	16 (5.0)
>400 NR:12	19 (5.9)
CD4+ level, cells/µL	
≤200	21(6.8)
201–500	74 (24.1)
>500 NR: 41	212 (69.1)
Currently taking antiretroviral therapy NR: 7	317 (96.4)
Time of antiretroviral therapy, years	
≤2	40 (12.8)
3–9	53 (17.0)
≥10 NR:22	219 (70.2)

Data presented as number (percentage, %) or mean (standard deviation, SD) and median (Interquartile range, IQR: p25; p75). NR, not reported, HIV, human immunodeficiency virus.

**Table 2 jpm-12-01778-t002:** Risk factors of HPV infection.

Characteristics	Values
Previous cervical cancer screening	297 (89.0)
HPV vaccination NR: 15	14 (4.4)
Previous HSIL	124 (37.1)
Vulvar	16 (4.8)
Vaginal	5 (1.4)
Cervical	81 (24.3)
Smoking	
Current Smoker	150 (46.7)
Former user	59 (18.4)
Never used NR: 13	112 (34.9)
Age of first sexual intercourse, years	
NR: 72	
Mean (SD)	17.8 (10.1)
Median (IQR)	17.0 (16.0;18.0)
Number of sexual partners in the last three years	
NR: 57	
Mean (SD)	1.1 (1.1)
Median (IQR)	1.1 (1.0;1.0)
Stable partner NR: 36	186 (62.4)
Anal sex NR: 101	59 (25.3)
Hormonal contraceptives NR: 85	
Current user	10 (4.0)
Former user	67 (26.9)
Never used	172 (69.1)
Barrier contraceptive user NR: 65	140 (52.0)
IUD user NR: 15	15 (4.7)
Previous STD NR: 14	134 (41.9)
HCV co-infection	102 (31.9)

Values expressed *n* (%), Mean (standard deviation, SD) and Median p50 (Interquartile Range, IQR, p25; p75) NR, no reported; HSIL, high grade squamous intraepithelial lesion; HPV, Human papilloma virus; IUD, intrauterine device; STD, sexually transmitted disease; HCV, Hepatitis C viral.

**Table 3 jpm-12-01778-t003:** Anal and Cervical screening results.

Characteristics	Location	
	Anal	Cervical
Cytology		
NILM, HPV not available	-	19 (5.7)
NILM, HPV negative	141 (42.2)	167 (50.0)
NILM, HR-HPV positive	61 (18.3)	49 (14.7)
ASCUS	107 (32.0)	54 (16.2)
ACG	0 (0.0)	1 (0.3)
L-SIL	17 (5.1)	30 (9.0)
ASC-H	3 (0.9)	3 (0.9)
H-SIL	4 (1.2)	6 (1.8)
Carcinoma	1 (0.3)	0 (0.0)
Not available	-	5 (1.5)
HPV test		
Negative	167 (50.0)	186 (61.6)
Positive, low risk	33 (9.9)	22 (7.3)
Positive, high risk	95 (28.4)	68 (22.5)
Positive, Low and High risk	39 (11.7)	26 (8.6)
Not available	-	32
HPV 16 positive	33 (9.9)	23 (6.9)
HRA	62 (18.6)	-
Negative	52 (83.9)	
G1	7 (11.3)	
G2	3 (4.8)	
Biopsy ^a^	12 (3.6)	34 (10.2)
Negative	4 (33.3)	9 (26.5)
LSIL	6 (50.0)	13 (38.2)
HSIL	2 (16.7)	12 (35.3)

Values expressed *n* (%) ^a^ LSIL, anal intraepithelial neoplasia type 1, or HSIL, anal intraepithelial neoplasia types 2/3. NILM, negative for intraepithelial lesion; ASCUS, atypical squamous cells of undetermined significance; ACG, Atypical glandular cells; ASC-H, atypical squamous cells, cannot rule out high-grade squamous intraepithelial lesion; LSIL, low-grade squamous intraepithelial lesion; HSIL, high-grade squamous intraepithelial lesion; HRA, high-resolution anoscope (HRA); HPV, Human papilloma virus; HR-HPV, High Risk of Human papilloma virus.

**Table 4 jpm-12-01778-t004:** Patient characteristics according to abnormal anal screening.

Characteristics	Abnormal Cytology			High Risk HPV Infection		
	No	Yes	*p*-Value	No	Yes	*p*-Value
Age, years			0.08			0.059
<35	29 (14.4)	9 (6.8)		16 (8.0)	22 (16.4)	
35–50	97 (48.0)	74 (56.1)		107 (53.5)	64 (47.8)	
>50	76 (37.6)	49 (37.1)		77 (38.5)	48 (35.8)	
Postmenopausal	84 (42.2)	57 (46.0)	0.5	83 (42.6)	58 (45.3)	0.626
HIV infection time, years			0.85			<0.001
≤2	19 (10.2)	15 (12.2)		12 (6.4)	22 (17.9)	
3–9	23 (12.3)	15 (12.2)		29 (10.6)	19 (15.5)	
≥10	145 (77.5)	93 (75.6)		156 (83.4)	82 (66.7)	
HIV viral load level, copies/mL			0.4			0.037
<50	175 (90.7)	112 (86.8)		176 (91.7)	111(85.4)	
50–400	9 (4.7)	7 (5.4)		10 (5.2)	6 (4.6)	
>400	9 (4.7)	10 (7.6)		6 (3.1)	13 (10.0)	
CD4+ level, copies/mL			0.28			0.002
≤200	10 (5.4)	11 (9.1)		7 (3.7)	14 (11.9)	
201–500	42 (22.6)	32 (26.5)		39 (20.6)	35 (29.6)	
≥500	134 (72.0)	78 (64.5)		143 (75.7)	69 (58.5)	
Current taking antiretroviral therapy	188 (96.4)	129 (97.7)	0.7 ^F^	193 (98.0)	124 (95.4)	0.204 ^F^
Time of antiretroviral therapy, years			0.652			0.017
≤2	24 (12.7)	16 (12.9)		16 (8.5)	24 (19.5)	
3–9	29 (15.4)	24 (19.4)		33 (17.5)	20 (16.3)	
≥10	135 (71.8)	84 (67.7)		140 (74.1)	79 (64.2)	
Smoking	122 (63.2)	87 (68.0)	0.3	126 (64.3)	83 (66.4)	0.698
Age of first sexual intercourse, years	18.4 (1.0)	16.9 (0.2)	0.249	18.3 (1.0)	17.1 (0.2)	0.3317
Number of sexual partners in the last three years	1.1 (0.1)	1.0(0.1)	0.38	0.9 (0.5)	1.2 (0.1)	0.17
Anal Sex	34 (23.8)	25 (27.8)	0.494	40 (20.2)	19 (20.9)	0.212
Previous STD	76 (39.4)	58 (45.7)	0.264	77 (39.7)	57 (45.2)	0.32
HCV co-infection	58 (28.7)	44 (33.3)	0.370	59 (29.5)	43 (32.1)	0.615
Previous cervical H-SIL	44 (21.8)	37 (28.0)	0.193	45 (22.5)	36 (36.9)	0.362
Current High risk HPV cervical infection	54 (28.4)	40 (35.7)	0.186	31 (17.4)	63 (50.8)	<0.001
HPV 16 cervical infection	14 (6.9)	9 (6.8)	0.968	11 (5.5)	12 (9.0)	0.222

Values expressed *n* (%) HIV, human immunodeficiency virus; STD, sexual transmitted disease; HCV, Hepatitis C viral; HPV, Human papilloma virus; IUD, intrauterine device; HSIL, high-grade squamous intraepithelial lesion; F, Fisher test.

**Table 5 jpm-12-01778-t005:** Univariate and multivariate analyses for high-risk HPV anal infection.

Characteristics	Univariate Analysis		Multivariate Analysis	
	0R (95% CI)	*p*-Value	0R (95% CI)	*p*-Value
HIV infection time, years	0.6 (0.38; 0.98)	<0.001	0.6 (0.28; 1.16)	0.124
HIV viral load level, copies/mL	0.5 (0.06;0.94)	0.026	1.2 (0.7; 2.13)	0.476
CD4+ level, copies/mL	0.7 (0.29; 0.97)	0.001	0.7 (0.41; 1.11)	0.127
Time of antiretroviral therapy, years	0.4 (0. 09; 0.72)	0.012	1.4 (0.7; 2.72)	0.346
Current High risk HPV cervical infection	1.6 (1.06; 2.11)	<0.001	5.1 (2.8; 9.18)	<0.001

HIV, human immunodeficiency virus; HPV, Human papilloma virus.

## 4. Discussion

The results of this study indicate that WLWHs have a high prevalence of HR-HPV anal infection; however, anal intraepithelial dysplasia is not a common finding among them. HR-HPV anal infection is more frequent in patients with a longer HIV infection, higher viral load level, lower CD4+ level, and shorter antiretroviral treatment time, and in those with HR-HPV cervical infection. Most importantly, HR-HPV cervical infection is the principal risk factor for HR-HPV anal infection (OR 5.1%, *p* < 0.001).

More than 90% of cases of anal cancer are associated with HR-HPV infection of the anal canal and/or perianal tissues [21]. Despite the strength of this association, no randomized clinical trials have documented the value of screening for anal squamous intraepithelial lesions (SILs) in at-risk populations. Instead, the rationale for screening relies on the similarities between the anus and cervix, the established success of cervical cytology screening in reducing the incidence of cervical cancer, and the demonstrated benefit of treating HSILs in reducing the rate of progression to invasive anal cancer [22].

The prevalence of HR-HPV anal infection has been described in the literature to be around 44% in the WLWHs population, with HPV 16 being the most frequently detected genotype [23,24,25,26,27]. In our study, we found that 40.1% of patients had HR-HPV anal infection, with a prevalence of 24.6% of HPV16 infection.

The incidence of abnormal anal cytology results varies in the literature from 2.1% to 74%, with a high-grade cytological result (including ASC-H, HSIL, and carcinoma) ranging from 0.5% to 10.7% [28,29,30]. The considerable difficulty in interpreting anal cytology and its low reproducibility could explain this wide range of findings [31,32]. We found a high prevalence of abnormal anal cytology (39.5%) with 2.4% of high-grade results. ASCUS was the most frequent result (32% of total samples, 81% of abnormal cytology), much more frequent than in cervical samples (16% in our study).

The mechanisms determining whether HSIL develops in the presence of HR-HPV infection are still incompletely understood. In our study, the prevalence of histological anal HSIL was 0.6% (16.7% of the biopsies performed), lower than what has been reported by other authors [33,34,35]. Gaisa et al. found the prevalence of histological anal H-SIL to be 5.1% (26% of the biopsied performed) in a total of 745 WLWHs [33]. As shown for cervical HPV infection, effective antiretroviral therapy, with higher levels of CD4 counts and lower levels of HIV viral load, reduces the prevalence of anal HR-HPV infection and HSIL in PLWH [35,36]. In our cohort, most of the women had a well-controlled immunological status and were undergoing active antiretroviral therapy (ART). The time of ART was more than 3 years in 87.2% of them: 89.1% had an undetectable HIV viral load and 93.2% had a CD4 count >200 cells/mm^3^. These characteristics could explain our low rate of anal HSIL. Additionally, some authors reported higher rates of H-SIL when performing random biopsies in patients with normal HRA [34]. In our study, biopsies were only performed in cases with abnormal HRA (12, 9.1%); thus, some lesions could have been missed, and the real prevalence could be under-rated.

Several studies have evaluated cervical HPV natural history and risk factors of acquisition, persistence, and precancer progression in WLWHs [7]. Conversely, little is still known about anal HPV natural history and risk factors for HPV anal infection. Palefsky et al. reported that HPV anal infection was associated with a CD4 cell count ≤200 cells/mm^3^, HPV cervical infection, and younger age (<36 vs. >45 years) [25]. Univariate analysis in our study showed higher rates of HR-HPV anal infection in patients with longer HIV infection, higher viral load level, lower CD4+ level, shorter antiretroviral treatment time, and in those with HR-HPV cervical infections. Age, smoking status, anal intercourse, or previous cervical HSIL were not associated with HR-HPV anal infection. The factor that remained significant in multivariate analysis in our study was HPV cervical infection (OR 5.1, *p* < 0.001), consistent with the literature. A high correlation between cervical and anal HPV infection has been reported, even with a higher prevalence of anal HPV infection than cervical HPV infection [37]. Transmissions between the cervical and anal areas may be one means of acquiring anal HPV infection, considering that no association was found between anal intercourse and anal HPV infection.

The main limitation of this study is its retrospective nature, which could have led to some missed confounding factors. Despite these limitations, our study represents a single-institutional cohort with a strict standardization of conditions for anal screening procedures, which was performed by a team specialized in HPV disorders. Additionally, tissue pathology was reported by a pathologist specialized in gynecological oncology in all cases, ensuring highly accurate reports in the majority of cases. Furthermore, to the best of our knowledge, this is the largest study on anal HR-HPV infection and the prevalence of anal abnormal cytology among WLWHs performed in Spain.

## 5. Conclusions

Our study showed a high prevalence of HR-HPV anal infection (40.1%) as well as a high prevalence of anal cytologic abnormalities (39.5%) in WLWH. Concomitant HR-HPV cervical infection is an independent risk factor for HR-HPV anal infection. The mechanisms influencing whether HSILs develop in the presence of HR-HPV infection are still incompletely understood; therefore, additional efforts should be made to determine the best screening strategies for this population.

## Figures and Tables

**Figure 1 jpm-12-01778-f001:**
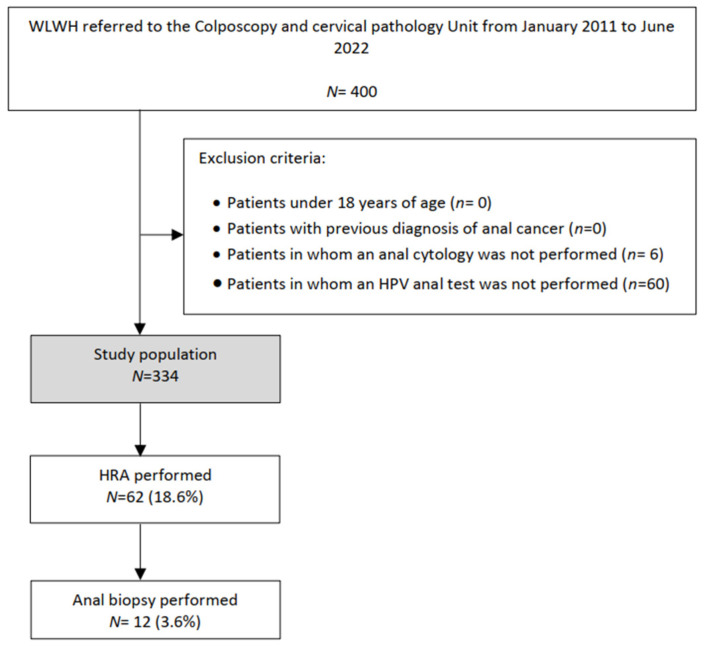
Flowchart of the study population: WLWHs, Women living with Human Immunodeficiency virus; HRA, high resolution anoscopy; HPV, Human Papilloma virus.

## Data Availability

The data presented in this study are available on request from the corresponding author. The data are not publicly available due to the decision of the authors.

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
