# Peer review of "Prevalence of Anal High-Risk Human Papilloma Virus Infection and Abnormal Anal Cytology among Women Living with HIV"

_jpm, 2022, doi:10.3390/jpm12111778_

Round 1
Reviewer 1 Report
The authors present a single-center observational cross-sectional study in women living with HIV who underwent anal cytology and anal HPV testing.
The study concluded that they have a high prevalence of anal HR-HPV infection and anal cytological abnormalities. In addition, they observed that cervical HR-HPV infection is the main predictor of anal HR-HPV infection.
The authors presented a study with clearly established results that could be used in future prospective and multicentre studies.
Author Response
Thank you very much for your comments and your recommendation for publication. Following your suggestions:
- We have made an English edition to correct the grammar and flow of the manuscript. We attach the edition certificate.
- We have improved the description of the methods (lines 54-55 and 65- 73 of the revised manuscript)
Reviewer 2 Report
The authors analyzed the prevalence of anal HR-HPV infection with abnormal anal cytology in a large cohort of women living with HIV. Additionally, they also assessed risk factors associated with anal HR-HPV infection. This reviewer found this to be a very interesting study and recommends it for publication pending minor revisions.
1. There are english spelling and grammar issues throughout the manuscript.
2. There are a bunch of acronyms that are not defined prior to their first use. Such as: HIV, OR etc.
3. Line 14 is missing CI after 95%.
4. The introduction is a bit light. Some more info on HIV and HPV would be appreciated. Especially highlighting the differences between HR and LR HPV types.
5. Line 64, authors state "serotypes of HPV". Different HPVs are not serotypes just types or genotypes.
Author Response
Response to Reviewer 2 Comments
Point 1: The authors analyzed the prevalence of anal HR-HPV infection with abnormal anal cytology in a large cohort of women living with HIV. Additionally, they also assessed risk factors associated with anal HR-HPV infection. This reviewer found this to be a very interesting study and recommends it for publication pending minor revisions.
Thank you very much for your comments and recommendations. We appreciate your acceptance of the manuscript for publication.
Point 2: There are english spelling and grammar issues throughout the manuscript.
We have made an English edition to correct the grammar and flow of the manuscript. We attach the edition certificate.
Point 3: There are a bunch of acronyms that are not defined prior to their first use. Such as: HIV, OR etc.
We thank the reviewer for this very important remark. All the acronyms have been defined prior to their first use and the list of abbreviations has been actualized.
Point 4: Line 14 is missing CI after 95%.
We have included CI after 95% in line 16 of the revised manuscript.
Point 5: The introduction is a bit light. Some more info on HIV and HPV would be appreciated. Especially highlighting the differences between HR and LR HPV types.
We have revised the introduction and we think that has been substantially improved (Lines 29 to 38 of the revised manuscript).
Point 6: Line 64, authors state "serotypes of HPV". Different HPVs are not serotypes just types or genotypes.
We have changed serotypes of HPV by genotypes (line 80 of the revised manuscript). We also thank the reviewer for this comment.
Reviewer 3 Report
Authors tried to correlate prevalence of HR-HPV among patients who have anal and cervical cacers. The conclusion reported the importance of using the cervical cancer as predictor for the anal cancer.
The comparison is nt clearly stated in the aim and design of the study. You know this as you go through the detail of the article or suddenly it appeared in the conclusion. A concomitant sampling from each/individul patients must be performed.
Whether the cervical cancer can be used as predictor for anal canacer, I would like to see if the cervical cancer lesions preceeded the anal cancer in comcomitant sampling as well. Although umber of cases of cervical cancer were much higher!
Accordingly the genotyping, particularly detecting the HR-HPV sould be less in number. Additionally, most literature stated that low risk-HPV were mosty dominant in anal cancer cases compared to cervical cancer where HR-HPV were dominant.
Author Response
Response to Reviewer 3 Comments
|
Point 1: Authors tried to correlate prevalence of HR-HPV among patients who have anal and cervical cacers. The conclusion reported the importance of using the cervical cancer as predictor for the anal cancer. The comparison is not clearly stated in the aim and design of the study. You know this as you go through the detail of the article or suddenly it appeared in the conclusion. A concomitant sampling from each/individul patients must be performed.
The main aim of our study was to analyze the prevalence of anal HR-HPV infection and the prevalence of abnormal anal cytology in a cohort of WLWH who were referred to Colposcopy and Cervical Pathology Unit of our Hospital for cervical cancer screening or follow-up for a history of cervical H-SIL. Additionally, we aimed to assess the risk factors for anal HR-HPV infection, including HR-HPV cervical infection, previous cervical H-SIL and other clinical features as age, time of HIV infection or HIV viral load level. (Lines 47 to 49 of revised manuscript). This study was not designed to correlate prevalence of HR-HPV among patients with anal and cervical cancer. Because of this, prior history of anal cancer was an exclusion criterion (Lines 55 to 56 of revised manuscript; figure 1 of revised manuscript). We have revised the methos section to clarify these aspects (Lines 53 -54 and 65-73 of the revised manuscript).
Point 2: Whether the cervical cancer can be used as predictor for anal cancer, I would like to see if the cervical cancer lesions preceded the anal cancer in concomitant sampling as well. Although number of cases of cervical cancer were much higher.
The results we show are based in the cytology and biopsies taken in concomitant sampling. In first visit, vaginal cytology, anal cytology, and HPV test were performed. In case of abnormal results, colposcopy and/or HRA were performed, and suspicious lesions were biopsied. We didn´t find any case of cervical cancer neither anal cancer in this cohort (table 3 of revised manuscript), but HSIL was more frequently diagnosed in cervical than anal samples (table 3 of revised manuscript). In univariate and multivariate analysis, current HR-HPV cervical infection was the only independent risk factor for HR-HPV anal infection (Line 134and table 5 of revised manuscript) We have revised and modify the methods section to clarify these aspects (Lines 54-55 and 65-73 of the revised manuscript).
Point 3: Accordingly, the genotyping, particularly detecting the HR-HPV should be less in number. Additionally, most literature stated that low risk-HPV were mostly dominant in anal cancer cases compared to cervical cancer where HR-HPV were dominant. |
As far as we have Know, the most frequent genotype detected in anal cancer is HPV 16. Relative contribution of HPV 16 to anal cancer is 87,2%, with very few cases in males detecting HPV 6 and 11 (de Sanjosé, et al. JNCI cancer Spectr 2018, 2: pky 045. doi: 10.1093/jncics/pky045). Other studies have shown a high prevalence of HR-HPV anal infection in WLWH as our study does, being HPV 16 the most detected genotype, not only in anal cancer but also in WLWH with anal HSIL or without abnormal anal cytology (Gupta, R et al. HIV Med 2022, 23:378-389. doi: 10.1111/hiv.13227; Goeieman, B.Jet al. J Acquir Immune Defic Syndr 2017, 75: e59-e64. doi: 10.1097/QAI.0000000000001300; Palefsky et al. J Infect Dis 2001, 183: 383-91. doi: 10.1086/318071; Heard, I et al. Clin Infect Dis 2015, 60: 1559-68. doi: 10.1093/cid/civ049)
We have commented and included all these references in our study.